# Evaluation and optimization of outdoor wind environment in block based on space syntax and CFD simulation

**Peng Cao\***, **Wenhui Li**

School of Architecture and Urban Planning, Lanzhou Jiaotong University, Lanzhou, Gansu Province, China

\* caopeng@mail.lzjtu.cn

## Abstract

The wind environment quality at the height of pedestrians can significantly affect the thermal comfort and physical and mental health of pedestrians, promote the diffusion of air pollutants and inhibit the formation of urban heat island effect, and has been paid more and more attention in the field of urban and rural planning. This paper takes Jianlan Road commercial pedestrian Street as an example to maximize the thermal comfort of pedestrians. Based on CFD numerical simulation technology and space syntax theory, the pedestrian wind environment of the accessible space of the block is selected for quantitative research. Through numerical simulation, the influence of block spatial form on the wind environment at pedestrian height under the initial condition of uniform air flow is analyzed, and some suggestions are put forward for the optimization of block spatial form. Finally, the block optimization scheme is verified and simulated. The visualization results show that the wind environment quality of the optimized high-accessibility space is significantly improved, the proportion of comfort zone is increased from 58.2% to 86%, and the static wind rate is reduced from 41.8% to 14%. The wind environment optimization effect is obvious.

**Data Availability Statement:** All the wind environment data files have been uploaded to the Figshare data repository: https://doi.org/10.6084/m9.figshare.25398250.

## 1. Introduction

Urban spatial form is an important factor affecting urban microclimate environment. Clark et al. found that urban form may play an important role in long-term air quality by studying the data of 111 urban areas in the United States. As early as 2001, Professor Adlophe of France proposed a series of urban form indicators corresponding to the study of urban microclimate. Around the 1980s, computer technology developed rapidly. Computer can numerically solve the urban spatial wind field through fluid dynamics equations, and visually and conveniently present the numerical simulation results of urban wind environment through visualization technology. With the development of time, a number of numerical simulation software such as Fluent, ENVI-met and PHOENICS have emerged. The research and application of Computational Fluid Dynamics (CFD) technology in urban wind environment is becoming more and more mature. Stathopoulos et al. [1] determined the wind speed and turbulence conditions between two building passages through wind tunnel measurements, and analyzed the

**Funding:** the study was funded by the Gansu Provincial Natural Science Foundation, but the funders had no role in study design, data collection and analysis, decision to publish, or preparation of the manuscript.

**Competing interests:** The authors have declared that no competing interests exist.

turbulence characteristics between buildings of different heights. In another study [2], he took the lead in simulating and analyzing the wind environment around seven parallel rectangular buildings in Ottawa through numerical simulation technology, and confirmed the agreement between the numerical simulation results and the wind tunnel tests. Chang C H et al. [3] used Fluent software and four different k-ε models to simulate and analyze the vortex situation in street canyons. Peter J et al. [4] use K-ε turbulence model to numerically simulate the outdoor wind environment of Queen Elizabeth II Square in Australia. Zhang A S et al. [5] simulated and analyzed the wind environment of a complex consisting of 18 buildings in a multi-layout scheme under different wind angles and different layouts. Cheng et al. [6] used computational fluid dynamics to simulate the street wind environment of residential regions, and conducted numerical simulation analysis on different height ratios of building groups. Jean Claus et al. [7] simulated and discussed the flow of different wind directions in the building array in a typical urban canopy area. C. Garcia-Sanchez et al. [8] used computational fluid dynamics to study airflow diffusion in urban buildings in Oklahoma, USA. M. Gloria Gomes et al. [9] simulated the wind pressure on the building surface of three building forms, namely cube, U shape and L shape, according to 8 different wind direction conditions.

In the research field of urban pedestrian-level wind, White B R [10] simulated the wind environment at the pedestrian height of San Francisco through wind tunnel experiment, proving the possibility of wind tunnel experiment in the study of urban scale wind environment. Chenyu Huang et al. [11] proposed a downscaling method and used WRF-CFD simulation technology to predict and evaluate urban pedestrian wind under 6 different configurations. The research shows that using the mesoscale data of 200 meters height as the parameters of CFD model is closer to the real situation of pedestrian wind in complex urban environments. Yueyang He, Abel Tablada et al. [12] have carried out wind environment evaluation on the connection mode of pedestrian space in a high-density urban central area through CFD numerical simulation. The result shows that under the condition of the same building density, adjusting the connection mode of pedestrian space can effectively optimize wind speed and pollutant concentration. A U Weerasuriya et al. [13] evaluated the effect of twisted wind on the wind field at pedestrian height through wind tunnel test, and the results showed that twisted wind would produce higher wind speed at the location with low density of nearby buildings. K T Tse et al. [14] simulated the wind environment around the " lift-up" buildings with nine different parameters, and the results showed that the core height of the " lift-up" buildings had the most significant influence on the area and size of the surrounding high wind speed zone and low wind speed zone. Campos [15] combined wind tunnel tests and CFD simulations to analyze the wind conditions of seven different shapes of 15-storey towers at the University of Nottingham. A complex urban model was developed to evaluate the comfort and safety of the wind environment at pedestrian height [16].

Space syntax is a theory and method to study the relationship between spatial organization and human society through the quantitative description of human settlements, including buildings, settlements, cities and even landscapes [17–21]. It was invented by Bill Hillier, Julienne Hanson and others at Barlett College, University of London. The theory of space syntax has been deeply involved in the careful study of the spatial essence and function of architecture and city, and has been constantly improved [22–26]. The result is a set of computer software that can be used for spatial analysis of the built environment at all scales. Space syntax is both a theory and a tool, and its three main tools include: J-Graph, Axial map/Segment map and Visibility Graph Analysis can be used to describe and analyze the influence of architectural space and urban space on people using quantitative indicators. It is widely used in the exploration of theory and practice. The integration is the most important parameter in space syntax theory, which reflecting the degree of closeness between a node and other nodes in the system [27–29]. The higher the

integration degree of node, it is easier for other nodes in the analysis range to reach this node. Take Wuhan as example, Li et al. [30] made a quantitative analysis of urban spatial organization and integration degree, and concluded that integration degree is a direct variable describing spatial accessibility. The greater the integration value, the higher the accessibility.

Space syntax theory and numerical simulation technology are both important achievements of urban research, and have been widely recognized and concerned in their respective fields. The combined research of the two will make the environment and space design have a real link. Therefore, this study is based on CFD numerical simulation and combined with space syntax to analyze the accessibility of the block space, so as to improve the pedestrian wind environment of the block. Firstly, DEPTHMAP and PHOENICS software are used to simulate the reachability and wind environment of the study area and locate the high reachability space to be optimized. Through numerical simulation, the influence law of building spatial form in the high accessibility space on the wind environment at pedestrian height was explored, and suggestions for optimizing the block height accessibility spatial form were put forward, so as to improve people's activity experience in the commercial pedestrian street area.

## 2. Study case analysis

The research area is Jianlan Road Commercial Pedestrian Street (Fig 1), which is located in Qilihe District, Lanzhou City. Due to the dense surrounding subway, bus and other transportation stations and convenient commercial facilities, and it have a large population density. This block undertakes important social, economic and cultural activities, and is one of the largest and most dynamic commercial centers in Qilihe District. This research scope is the part of Jianlan Road Commercial Pedestrian Street, which is located at the intersection of Xijin West Road and Dunhuang Road, east to Jianlan Road, west to Shizi subway Station, south to Xijin West Road, and north to Yangguang Homeland. The total area is about 9.4 hectares.

## 3. Methods

### 3.1. Meteorological data processing

To ensure more accurate meteorological data, this study used the ladybug plugin to link to China Standard Weather Database (CSWD) and obtain *epw* format file data. CSWD data is sourced from China Meteorological Administration, including 8,760 hours of measured meteorological data, including temperature, wind speed, wind direction and solar movement trajectory throughout the year, so that accurate hourly data can be obtained.

In the actual state, the urban wind environment belongs to the dynamic wind, which has the dual dimension attribute of time dimension and space dimension. Therefore, this study uses seasonal dominant wind direction and average wind speed as the basis of wind environment research to transform dynamic wind into steady wind. In recent years, the phenomenon of heat island in Lanzhou is obvious. Li Y et al. [31] conducted regression analysis of urbanization and climate change in Lanzhou by using the method of comparison between cities and suburbs. The results show that in the past 40 years, with the intensification of urbanization and the increase of population in Lanzhou, the effect of urban heat island has increased significantly, and the phenomenon of heat island in summer is the most intense. In addition, relevant studies have shown that the urban heat island effect is negatively correlated with the urban ventilation capacity, so this study uses the wind speed and wind direction data in summer when determining the wind direction and wind speed simulated for the wind environment of buildings. ladybug was used to visualize and analyze the annual meteorological data of Lanzhou city, and the dominant wind direction and average wind speed in June, July and August were used as the simulation data of the wind environment of Lanzhou City in summer months (Fig 2).

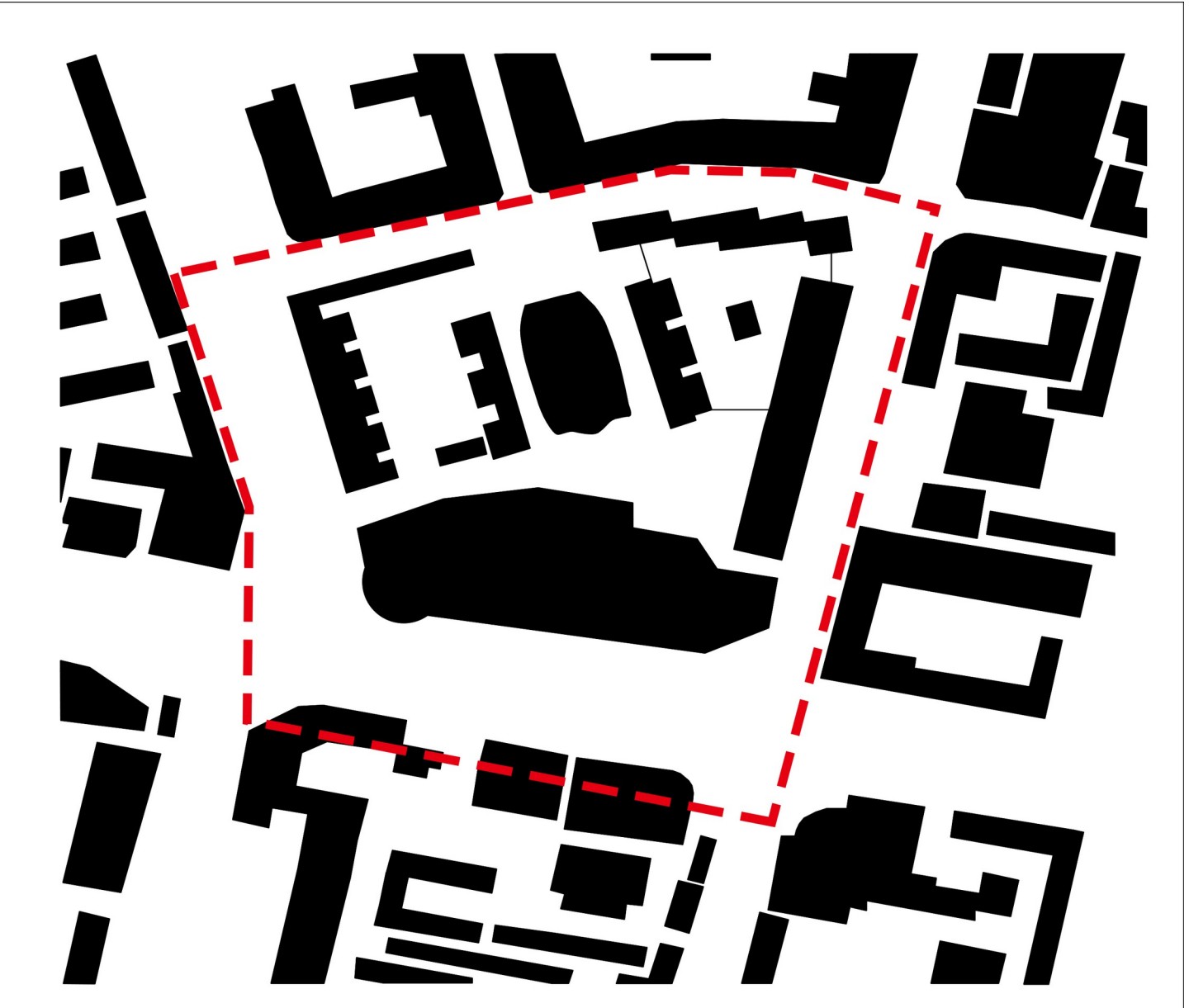

**Fig 1. Research scope of pedestrian commercial block on Jianlan Road (Author's self-painting).**

Through ladybug plug-in, the meteorological data of wind direction and wind speed in Lanzhou from June to August are screened and processed. It can be seen that the wind direction in Lanzhou in summer is mainly between north and northeast, and the dominant wind direction is NNE wind. Further accurate processing of its data resulted the average wind speed is 1.21m/s. Based on commonly used wind direction calculation methods, the formula for calculating wind direction is

$$D_{\mathrm{avg}} = \sum\nolimits_{i=1}^{n} D_i / n \tag{1}$$

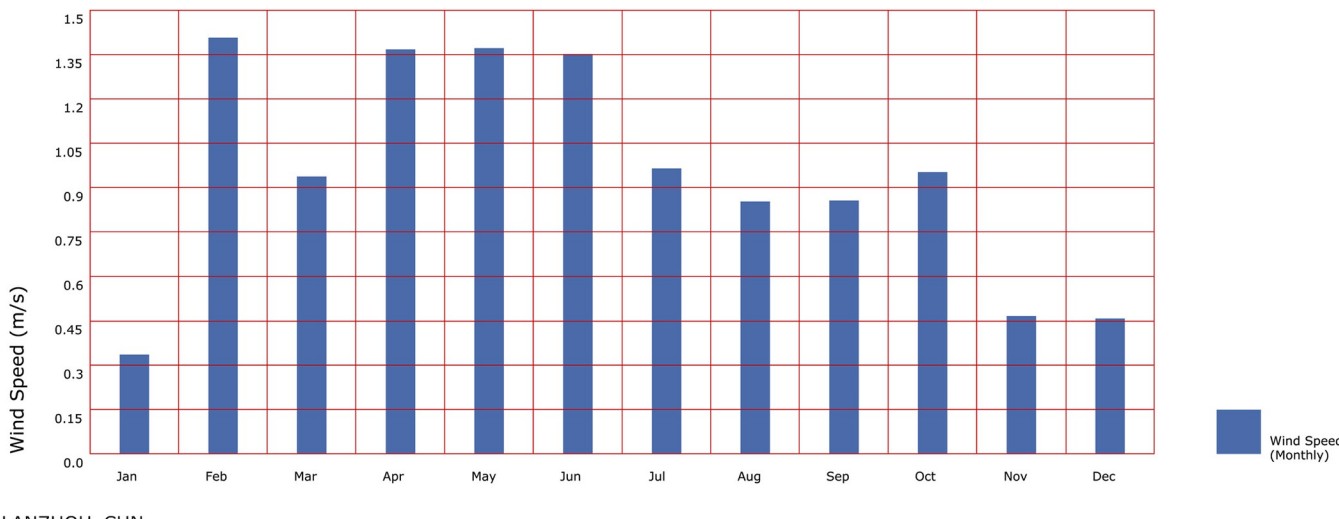

LANZHOU_CHN

**Fig 2. Average wind speed of monthly in Lanzhou.**

In the formula, $D_{avg}$ is the average wind direction, $D_i$ is the wind direction azimuth angle of $i$ wind direction samples, and $n$ is the number of samples.

The average summer wind direction is calculated is 27.5° northeast, which serves as the boundary reference data for PHOENICS wind environment simulation.

## 3.2. CFD numerical model establishment

Combined with the obtained building contours and field research results, a 3-dimensional model of the block was established. In the modeling process, on the premise of not affecting the airflow distribution around the building, the outline details of the building were properly simplified. Because the blocking effect of the buildings around the block can also affect the wind environment of the study region, in order to close to the real situation, modeling was also carried out on some of the surrounding buildings.

**3.2.1. Selection of mathematical model.** A variety of fluid calculation mathematical models are available in PHOENICS software. This study simulates the distribution of wind environment flow field at a height of 1.5 m, and the air flow of the outdoor pedestrian height is incompressible low-speed turbulence, so the standard $k$-$\varepsilon$ model is selected for simulation. This turbulence model approximates the first order closure of Reynolds stress, and its control equation is expressed by Reynolds mean N-S equation:

$$\frac{\partial U_i}{\partial \mathcal{X}_i} = 0 \tag{2}$$

$$U_j \frac{\partial_{U_i}}{\partial_{\mathcal{X}_j}} = -\frac{1}{\rho}\frac{\partial P}{\partial x_i} + \frac{\partial}{\partial x_j}\left(\mu \frac{\partial_{U_i}}{\partial_{\mathcal{X}_j}}\right) + \frac{\partial}{\partial x_j}\left(\overline{-\mathcal{U}_i\mathcal{U}_j}\right) \tag{3}$$

$$\overline{-\mathcal{U}_i\mathcal{U}_j} = 2C_\mu k s_{ij} - \frac{2}{3}k s \delta_{ij} \tag{4}$$

In the formula, $U$ represents the average velocity component, $x_i$ and $x_j$ represent the coordinate direction, $-\overline{\mathcal{U}_i\mathcal{U}_j}$ is the Reynolds stress, $P$ is the average pressure, $\rho$ is the air density, $\delta_{ij}$ is the Kronecker function, and the constant $C_\mu = 0.087$.

According to the classical turbulence theory, the expressions of turbulent kinetic energy and dissipation rate are:

$$q_{\text{sgs}} = \int_{k_c}^{\infty} E(k)dk \tag{5}$$

$$\varepsilon = 2v \int_{0}^{\infty} k^2 E(k)dk \tag{6}$$

**3.2.2. Determine the calculation region.**   During the software simulation process, the delineation of the calculation region will significantly affect the accuracy of the simulation results. If the calculation region is too small, it will be inconsistent with the flow field under realistic conditions. If the calculation region is too large, it will lead to long cycle and increase cost. Reasonable delineation of the calculation region is very important for the simulation results. According to "*Green Building Evaluation Standards*", the calculation region of this study is set be centered around the target building, with a vertical calculation boundary greater than 3H; The calculated boundary in the horizontal direction is greater than 5H (H is the building height value of the boundary in the calculated region). The model is finally placed in the calculation region (Fig 3) of 2100m×1920m×348m by calculating the range of blocks and surrounding buildings.

**3.2.3. Computational grid division.**   Meshing refers to each cell network that is calculated within the computation area, that is, within the rectangle. The size of the grid is directly related to the authenticity of the simulation results. If the number of grids is large, the grid is too dense, which will lead to the computer simulation running time is too long. The simulation results are not accurate enough because the number of grids is small and the grid is sparse.

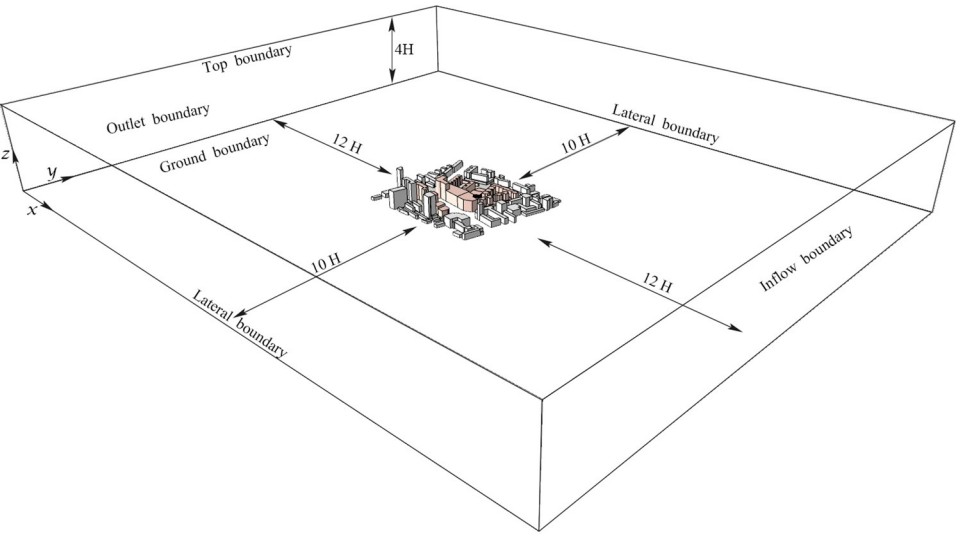

**Fig 3. The computational region.**

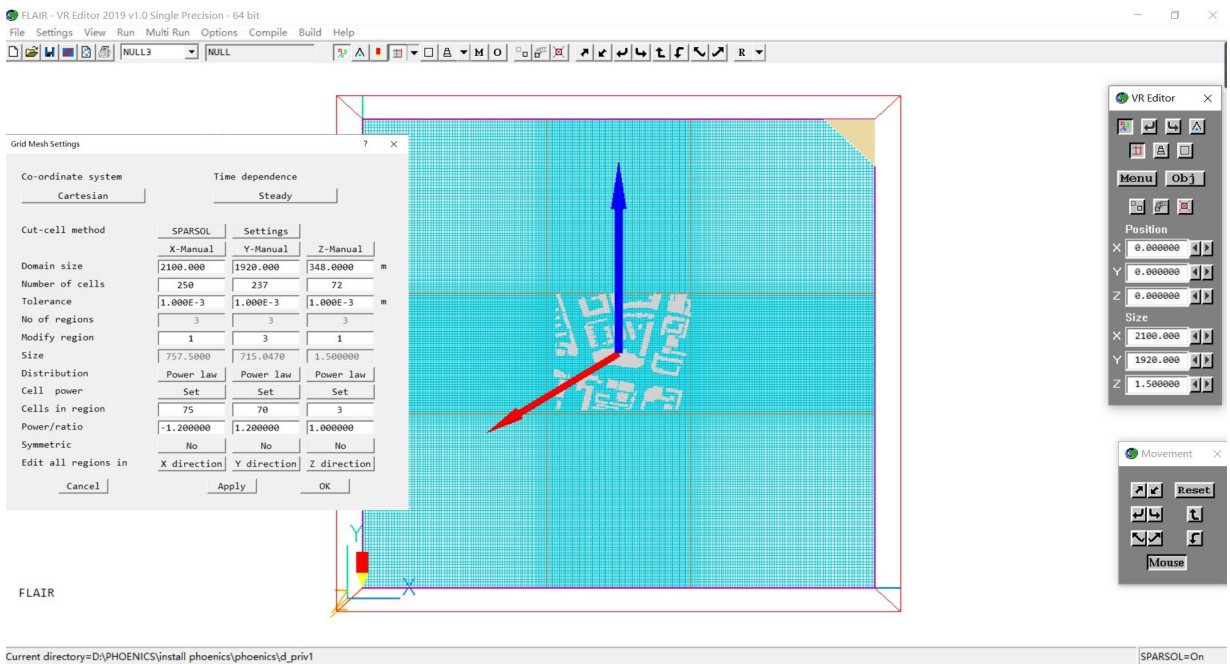

**Fig 4. The grid setting of X, Y, and Z axis of the Jianlan Road Pedestrian Commercial District.**

Therefore, the boundary of the model needs to be considered in many aspects in the reasonable setting of grid division.

As this paper focuses on the wind environment at the height of pedestrians, we adopt a locally encrypted grid division method for the grid division within 1.5m pedestrian height, and adopt a rectangular structured grid in the core building area here, and ensure that the size ratio of two adjacent grids is less than or equal to 1.2. In the edge area of the computing domain, the grid division is gradually sparse and the number of grids is gradually reduced. A composite grid method is adopted as a whole to reduce unnecessary computing processes (Fig 4).

**3.2.4. Set boundary conditions.** In the wind environment simulation, in order to get close to the real atmosphere and ensure the authenticity and accuracy of the calculation, it is necessary to consider the inlet boundary conditions and ground boundary conditions. This paper studies the wind environment at a height of 1.5 m, so the inlet boundary is analyzed using exponential relation and modified gradient wind. The ground boundary mainly considers the influence of roughness on the wind environment. According to the classification in the Table 1, the commercial pedestrian street belongs to Class C terrain, so the roughness value is 0.22.

For the boundary conditions of the entrance and exit, the wind speed of the entrance flow is uniformly distributed. Different heights produce different wind speeds, and the wind speed

**Table 1. Comparison of wind environment quality in high reachability space before and after optimization.**

| Wind environmental quality | Average wind speed (m/s) | High accessibility space area (m²) | Comfort zone area (m²) | Comfort zone ratio (%) | Non-comfort zone area (m²) | Non-comfort zone ratio (%) |
|---|---|---|---|---|---|---|
| Before optimization | 0.32~0.39 | 29130.69 | 11127.92 | 38.2 | 18002.77 | 61.8 |
| After optimization | 0.60~0.68 | 29130.69 | 25052.39 | 86 | 4078.30 | 14 |

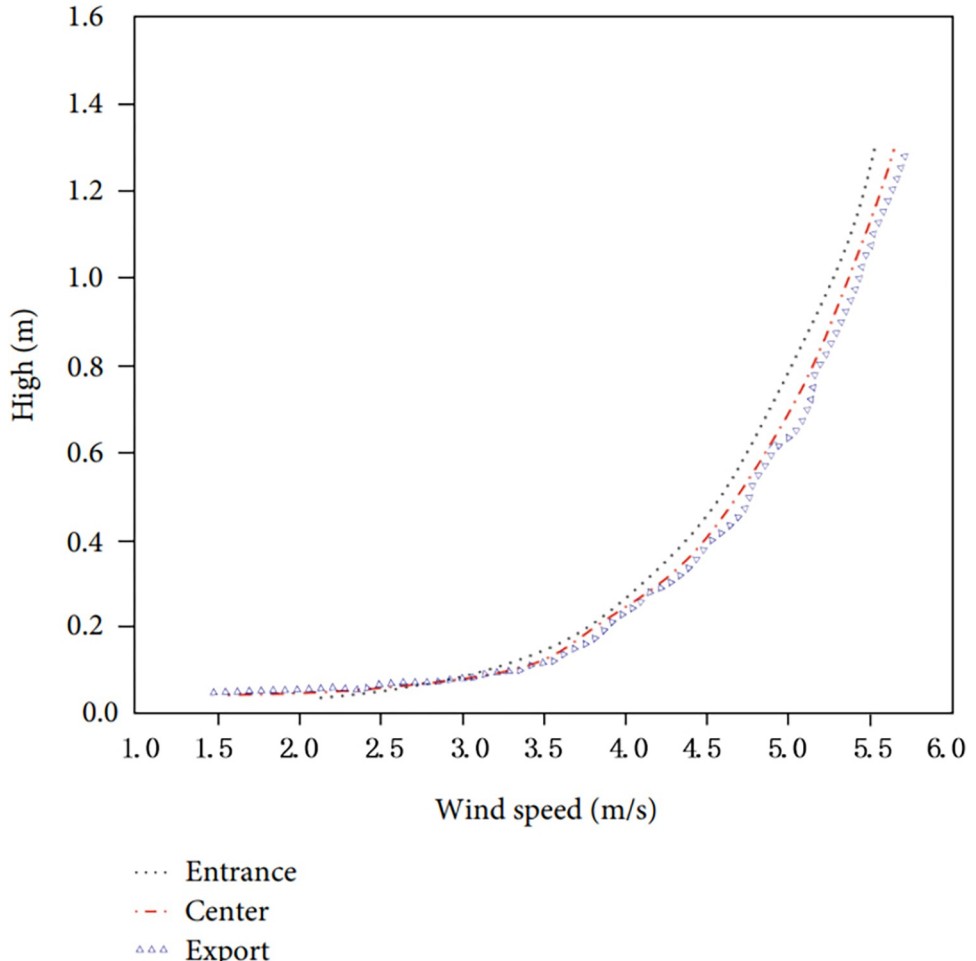

**Fig 5. Wind speed at the inlet, center, and outlet profiles of the CFD computational domain.**

increases with height. The formula for calculating the altitude and wind speed is as follows:

$$\mathcal{V}_h = \mathcal{V}_0 \left(\frac{h}{h_0}\right)^n \tag{7}$$

Where $\mathcal{V}_h$ is the wind speed at height $h$, in m/s; $\mathcal{V}_0$ is the wind speed at the reference height $h_0$, in m/s, generally taken as the wind speed at 10 m; and $n$ is the exponent, taken as 0.22.

Horizontal Homogeneity Test of the Flow Field. The atmospheric boundary layer should maintain horizontal homogeneity between upstream and downstream, so the inflow characteristics of computational fluid dynamics (CFD) simulations should be stable throughout the computational domain. In this study, the CFD numerical simulation of the wind tunnel test flow was carried out in a blank computational domain, and the wind speed flow characteristics at the center were extracted and compared with the inflow characteristics. As shown in Fig 5, the CFD model established in this study basically meets the requirements of horizontal homogeneity. The wind speed characteristic curves of the inlet, center, and outlet are basically consistent.

**3.2.5. Analysis of simulation results.** Based on PHOENICS software, the wind speed simulation results at the pedestrian height of 1.5 m in the block are obtained, as shown in the

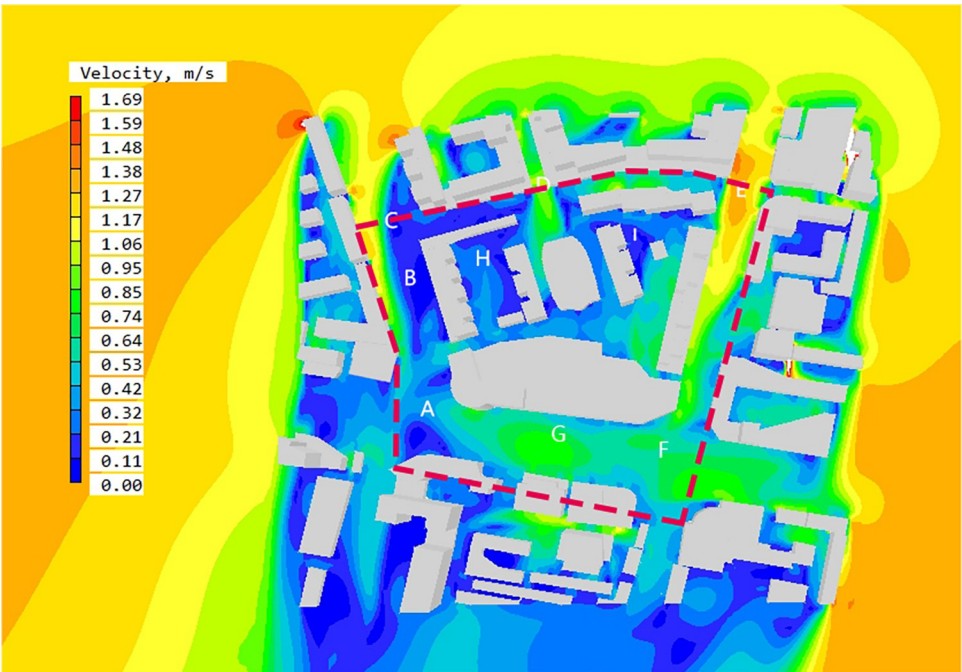

**Fig 6. The wind velocity contour map at Z = 1.5 m in summer.**

Fig 6. The average wind speed at the height of 1.5 m in the block is1.16m/s. From the figure, it can be seen that there is no strong wind area in the block, but there are obvious clam wind areas in some spaces. In order to further analyze the wind speed situation in the block, the result diagram is enlarged and some spatial nodes are numbered, and the causes are analyzed by combining the wind direction streamline diagram.

According to the wind speed chart, as for the overall block, the wind speed in space A,D, E, F and G spaces is relatively high. Combined with the wind direction analysis chart and the surrounding building layout, this paper summarizes the reasons for the formation of high wind speed space in the block as follows:

Wide street openings can bring strong winds. A,D, E, F and G region are located at street intersections, and the street entrance openings are wide. There is a large amount of northeast wind introduced into the interior of the street and converged at the intersections. The space between the two buildings on the south side of area D is relatively narrow, and the street boundary faces inward depression. This kind of space form is easy to form drafts, resulting in *the effect of narrow*, which makes local space wind speed increase rapidly.

Local circulation will be formed due to the partial enclosure between the buildings. The wind direction condition set before the software simulation is 27.5˚east of north. Region G is located in the middle of the street. The wind direction of the air flowing into the street changes due to the obstruction of buildings, forming local circulation and making the wind speed there higher than the surrounding area.

The low continuity of the street can introduce wind flow. The overall wind environment of E, F, and the surrounding streets regions is relatively good, and there is no obvious static wind region. Due to the obvious passage spacing in the leeward buildings on the east side of the street, the strong closure of the street is weakened, so the northeast wind can continue to flow into the street and run through the whole block.

Except for the region with relatively good wind speed in the block, the wind speed in most public spaces within the block has not reached the evaluation index, and there are large regions of static wind in spaces B, C and I. Combining with the wind direction analysis chart and spatial layout form, this paper makes the following analysis:

The building is highly enclosed. the west street of B and C region is close to perpendicular to the dominant wind direction, and the continuity of street buildings is too high, and there is no good outlet, resulting in most of the incoming wind is blocked by buildings, resulting in poor liquidity of the whole street and low wind environment quality.

The multi-direction wind flow is weakened due to mutual impact, forming a local wind speed reduction zone. The direction of air flow around the space I is chaotic, and the wind speed is weakened due to the impact of air flow in different directions, resulting in local vortex formation and static wind area.

## 3.3. Research on block accessibility based on space syntax

The space syntax theory has high reliability in evaluating space accessibility, and the research method based on topological network is more suitable for the reachability analysis of street scale. Therefore, this paper selects DEPTHMAP 1.0 software to analyze the view grid of street space, and parameters such as the visual integration of blocks are used as the measurement index of reachability.

Firstly, the outline of the building is obtained from the map and extracted in CAD to obtain the base map of space analysis. The files were converted into DXF format and imported into DEPTHMAP software for integration analysis. The results were imported into ArcGIS for space interpolation visualization of integration value, and the street sight integration analysis chart was obtained.

Before the analysis of the building space, the grid needs to be divided and the analysis precision determined. Considering the scale of the human body and the building space, and the commercial block is taken as the research unit in this study, the grid in DEPTHMAP is set at 2m×2m (Fig 7), which is sufficient for this study.

**3.3.1. Result analysis.**   The block view grid analysis was carried out to obtain the block integration map (Fig 8). The integration analysis map used color to describe the integration range value. The warmer the color (red), the higher the integration degree, and the colder the color (blue), the lower the integration degree. According to the results of integration analysis, the relative distribution of reachability within a block can be found intuitively. Nodes with relatively high integration are located at the intersections of several paths outside the block, with a larger scope of view acquired and a higher reachability. Nodes with low integration degree are mainly located in the residential space enclosed within the block. To make the division of high reachability region more accurate, this study divides the node region with integration value greater than 10 into high reachability space.

# 4. Analysis of outdoor high accessibility space wind environment

## 4.1. Analysis of wind environment in high reachability space

The PHOENICS software is used to simulate the outdoor wind environment at the pedestrian height of the block, and to obtain the wind speed distribution diagram and wind direction analysis diagram. Then, the DEPTHMAP software is used to obtain the block visual integration degree result distribution diagram to define the block height reachability space. The wind speed and wind direction analysis diagram are superimposed with the integration result. Wind speed and direction analysis diagram of high reachability space is obtained (Fig 9).

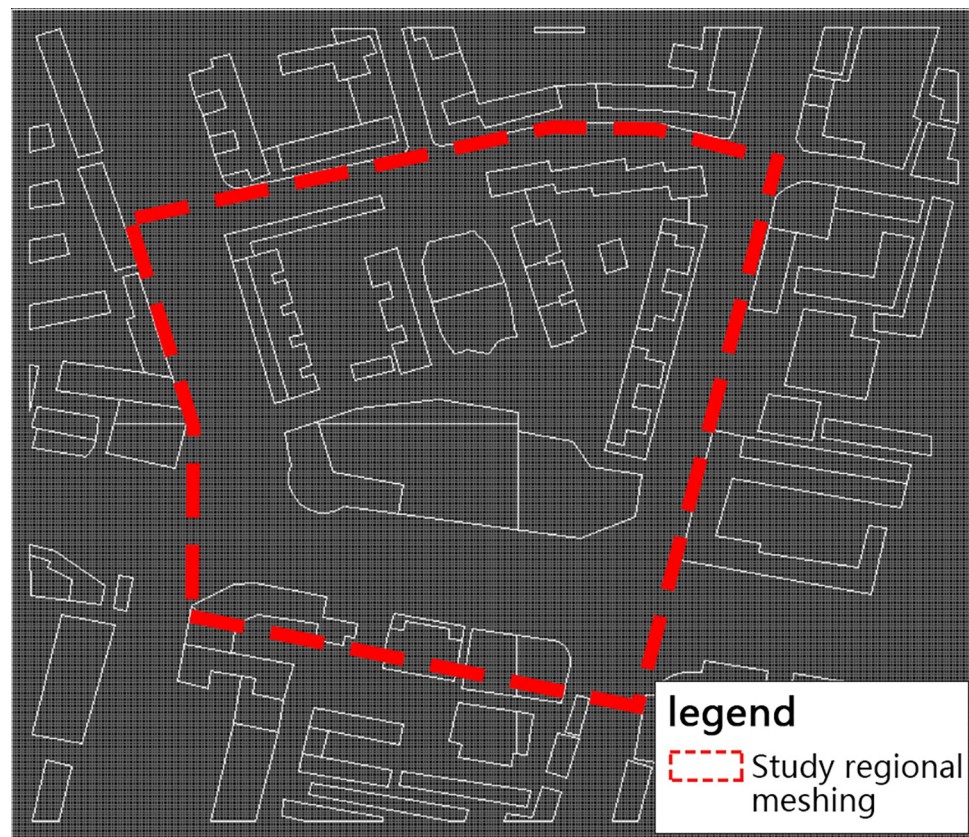

**Fig 7. The grid setting of DEPTHMAP.**

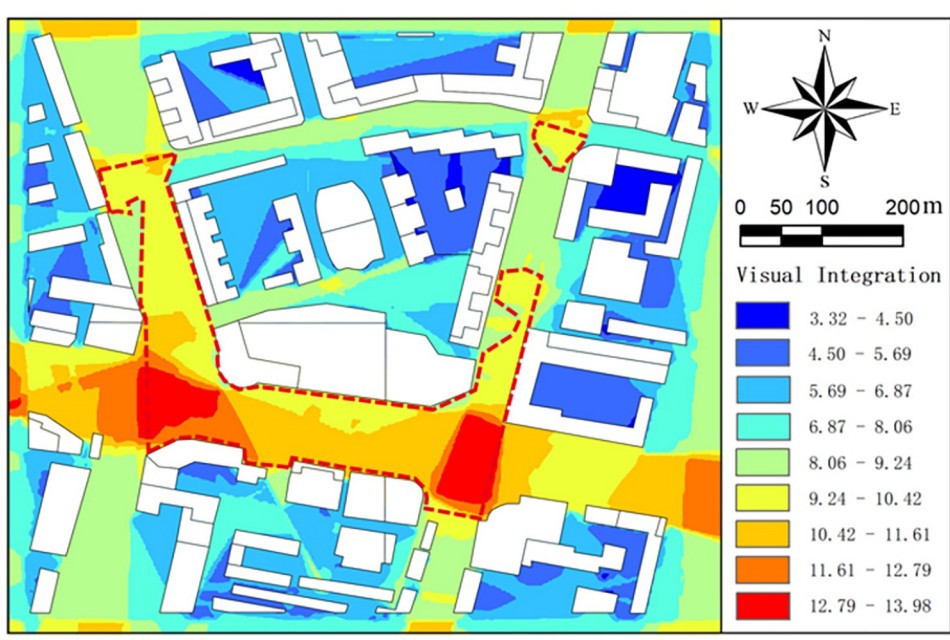

**Fig 8. The block integration map.**

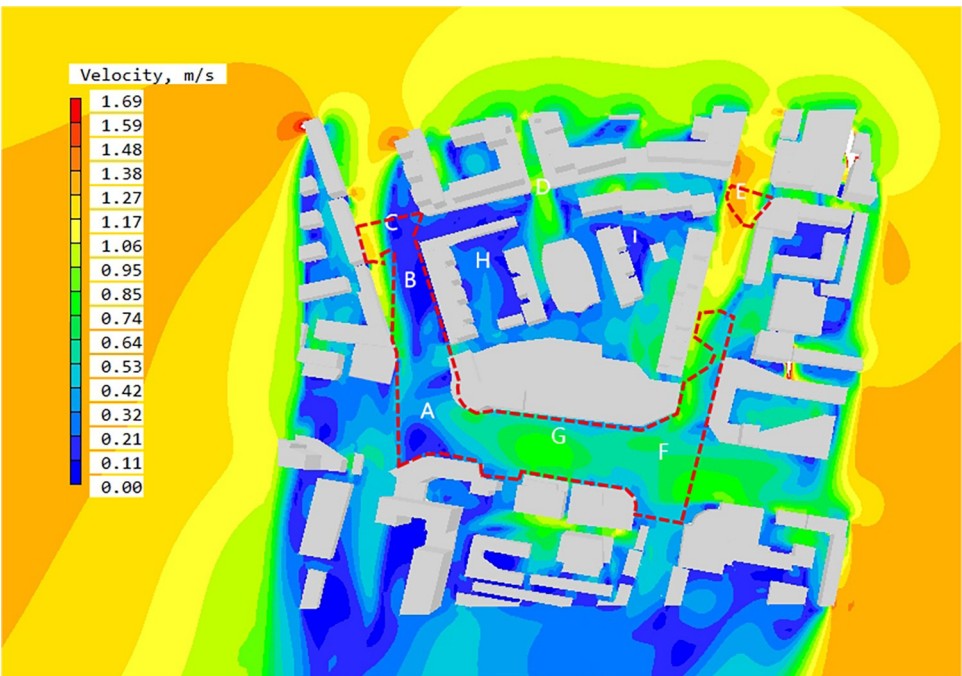

**Fig 9. The wind speed contour map at Z = 1.5m in high accessibility space in summer.**

By interpreting the wind speed and direction diagram of high reachability space, the following results can be obtained: ① The average wind speed in high reachability space is 0.53~0.64 m/s, the comfortable region with wind speed in 1.0~5.0 m/s accounts for 38.2% of the area of high reachability space, and the region with wind speed below the standard accounts for 61.8% of the area of high reachability space. ② The high reachability region is mainly located at the intersection of urban arterial road and street. A, B, C, E, F and G are located in the high reachability region. Among the five spaces, the wind speed in A, E, F and G is relatively high, with the average wind speed reaching 1.06m/s. Space B and C have obvious static wind regions, which the static wind rate reaching 80%.

The strong wind area ratio RQ is defined as

$$R_Q = \left(\frac{S_Q}{S_0}\right) \times 100\% \tag{8}$$

where $S_Q$ is the area occupied by the strong wind area in the flow field (m$^2$) and $S_0$ is the area (m$^2$) at the height of the strong wind area without interference from buildings.

In the same way, the quiet wind area is the area with a small value of urban spatial wind speed. In the urban space, it is a quiet wind area. Through the size of the area ratio index of the quiet wind area, the proportion of the poor wind environment can be judged. The quiet wind area ratio RJ is defined as

$$R_J = \left(\frac{S_J}{S_0}\right) \times 100\% \tag{9}$$

In the formula, $S_J$ is the area occupied by the calm wind area in the flow field (m$^2$) and $S_0$ is the area (m$^2$) at the height of the calm wind area when it is not disturbed by buildings. Based on the postprocessing of the simulation results of Airpak software, Photoshop software was

used to process the air age cloud map; that is, the air age values and areas represented by different color blocks were calculated, and the weighted average method was used to calculate the average air age of the study area. The mean air age A is defined a

$$\overline{A} = \frac{A_1 \times S_1 + A_2 \times S_2 + \cdots A_n \times S_n}{S_{1+} S_{2+} \ldots S_n} \tag{10}$$

In the formula, A is the average air age (s), An is the air age represented by different color blocks, and $S_n$ is the time spent in $A_n$. Combined with the practical application of this paper, the middle wind speed ratio is used as the relative comfort zone wind speed ratio $R_S$ to judge the size of the area at the height of 1.5 m that meets the comfort of outdoor pedestrians. The relative comfort zone wind speed ratio $R_S$ is defined as

$$R_S = 1 - R_Q - R_J \tag{11}$$

## 4.2. Suggestions on optimization of pedestrian wind environment in high accessibility space

By analyzing the simulation results of wind speed and wind direction for block (Fig 10), aiming at the wind environment problems in internal high-accessible space, measures such as staggered layout and adjustment of street opening interface are proposed for accurate optimization (Fig 11). After optimization, the static wind rate of high reachability space is reduced to 14%, the average wind speed reaches 0.85~1.06 m/s, and the overall average wind speed of the block is increased to 1.16 m/s (Table 1). The following summarizes the optimization measures for different regions.

1. Adjust the street intersection building opening form. Area C is located near the T-intersection, and its accessibility is relatively high. The corner building form is a right Angle, which blocks the southward flow of the northeast wind flowing in along the street, so a local quiet

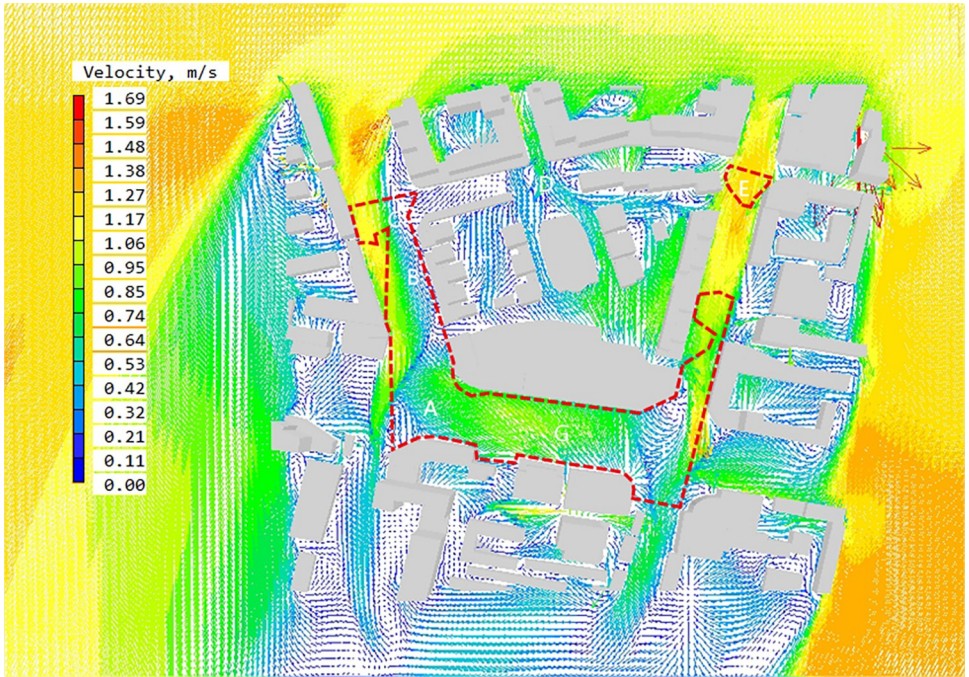

**Fig 10. The wind velocity vector illustration at Z = 1.5 m in high accessible space in summer after optimization.**

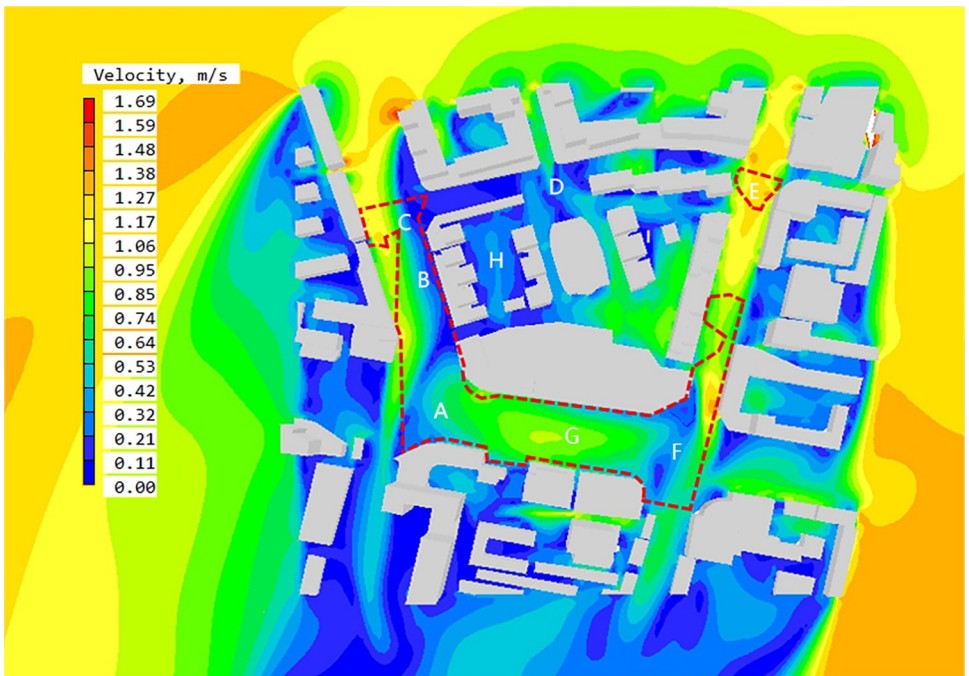

**Fig 11. The wind speed contour map at Z = 1.5 m in high accessible space in summer after optimization.**

wind area is formed below the intersection. To solve this problem, the optimization measures are applied to oblate the right-angle building and guide the wind flow to the south. As can be seen from the wind direction analysis diagram, the northeast wind at the entrance of the street at C collects with the wind flow above the street, causing a large amount of wind flow to flood into Dunhuang Road on the west side of the outer street. The overall wind environment of the street on the west side is improved, and the quiet wind area of the street is reduced from 67% to 33%.

2. Adjust building height. As can be seen from the above analysis of the pedestrian wind environment caused by changes in the height of residential buildings, the area of the wind shadow area on the lee side of the building shows an obvious increasing trend with the increase in building height. Therefore, in order to reduce the area of the wind shadow area in space B, the height of residential buildings is reduced from the original 36m to 30m within a reasonable adjustment range. It can be seen from the optimized wind speed contour map that the area of wind shadow in space B is greatly reduced after the height of residential buildings is reduced.

3. Reduce street continuity. Due to its high enclosing degree, the high-continuity street has a greater barrier effect on air flow. Therefore, in order to improve the pedestrian wind environment of the street on the west side of the block, the building interval between the residential buildings on the west side is increased to form vents, guide the flow of northeast wind on the west side of the street, and increase the overall wind speed.

## 5. Conclusion and outlook

This study attempts to improve the wind environment in the high reachability area through the wind environment simulation and reachability analysis of the Jianlan Road Commercial

Pedestrian Street in Lanzhou City. Through analysis of wind speed and direction analysis diagram, the wind environment in the highly reachability space is accurately optimized. On the premise of not making significant changes to the original planning layout as far as possible, the optimization scheme was finally determined after several simulation verifications. The following main conclusions were obtained from the simulation of this study:

1. Wide street openings can introduce large air currents. According to the wind speed analysis chart, it can be clearly seen that the region with wider streets can introduce large air flow, and its wind speed is obviously higher than that of narrow streets. Moreover, the direction of the main road should follow the dominant wind direction as far as possible.

2. Enclosed layouts and areas with high building continuity tend to form a "wind wall effect" that impedes airflow. For high-rise buildings with linear layout, it is advisable to avoid vertical arrangement with the dominant wind direction, and staggered layout can be used to appropriately improve the building's air permeability.

3. The multi-direction air flow will be weakened by mutual impact, forming a local wind speed weakening zone will be formed. A large opening space is beneficial for wind speed flow, but it may not necessarily guarantee the comfort of the wind environment.

4. The change of architectural spatial form will significantly affect the quality of pedestrian wind environment. Through the analysis, it is found that the change of building length, height and orientation will have an impact on the pedestrian wind environment, among which the change of building height has the greatest impact. With the increase of building height, the area of wind shadow area on the lee side of the building shows an obvious increasing trend.

This study only focuses on improving the wind environment of a block by combining space syntax and CFD numerical simulation. Due to limited research time and conditions, the precise optimization strategy proposed in this paper is basically based on experience and respect for the original planning and layout, without exploring the possibility of further research measures. In future optimization studies, it can be attempted to combine multiple research methods to analyze and optimize the wind environment, compare multiple quantitative methods and select the best to improve research rigor.

## Acknowledgments

We thank the China Standard Weather Database (CSWD) for providing the meteorological data. The helpful comments and suggestions from the all reviewers are very gratefully acknowledged.

## Author Contributions

**Conceptualization:** Peng Cao, Wenhui Li.

**Data curation:** Wenhui Li.

**Formal analysis:** Wenhui Li.

**Funding acquisition:** Wenhui Li.

**Investigation:** Wenhui Li.

**Methodology:** Wenhui Li.

**Project administration:** Wenhui Li.

**Resources:** Wenhui Li.

**Software:** Wenhui Li.

**Supervision:** Wenhui Li.

**Validation:** Peng Cao, Wenhui Li.

**Visualization:** Wenhui Li.

**Writing – original draft:** Wenhui Li.

**Writing – review & editing:** Wenhui Li.

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
