## [Decision Letter · Decision Letter 0]

10 Jul 2023

PONE-D-23-16872Evaluation and optimization of outdoor wind environment in block based on space syntax and CFD simulationPLOS ONE

Dear Dr. Li,

Thank you for submitting your manuscript to PLOS ONE. After careful consideration, we feel that it has merit but does not fully meet PLOS ONE’s publication criteria as it currently stands. Therefore, we invite you to submit a revised version of the manuscript that addresses the points raised during the review process.

Based on the reviewers suggestions, I have decided that your paper could be considered for publication if it is revised substantially in accordance with the reviewers' comments.  

Therefore, we invite you to submit a revised version of the manuscript that addresses the points raised during the review process.

We look forward to receiving your revised manuscript.

Kind regards,

Muhammad Shakaib, PhD

Academic Editor

PLOS ONE

Journal Requirements:

"NO - Include this sentence at the end of your statement: The funders had no role in study design, data collection and analysis, decision to publish, or preparation of the manuscript."

5. We note that Figures 1, 13, 14, 15, 16, 20, and 21 in your submission contain [map/satellite] images which may be copyrighted. All PLOS content is published under the Creative Commons Attribution License (CC BY 4.0), which means that the manuscript, images, and Supporting Information files will be freely available online, and any third party is permitted to access, download, copy, distribute, and use these materials in any way, even commercially, with proper attribution. For these reasons, we cannot publish previously copyrighted maps or satellite images created using proprietary data, such as Google software (Google Maps, Street View, and Earth). For more information, see our copyright guidelines: http://journals.plos.org/plosone/s/licenses-and-copyright.

a. You may seek permission from the original copyright holder of Figures 1, 13, 14, 15, 16, 20, and 21 to publish the content specifically under the CC BY 4.0 license.  

Reviewers' comments:

Reviewer's Responses to Questions

**Comments to the Author**

1. Is the manuscript technically sound, and do the data support the conclusions?

Reviewer #1: Yes

Reviewer #2: Partly

Reviewer #3: Partly

2. Has the statistical analysis been performed appropriately and rigorously? 

Reviewer #1: N/A

Reviewer #2: No

Reviewer #3: No

3. Have the authors made all data underlying the findings in their manuscript fully available?

Reviewer #1: Yes

Reviewer #2: No

Reviewer #3: Yes

4. Is the manuscript presented in an intelligible fashion and written in standard English?

Reviewer #1: Yes

Reviewer #2: Yes

Reviewer #3: Yes

5. Review Comments to the Author

Reviewer #1: The paper is an interesting study and a timely research work, evaluating and optimising outdoor wind environment in block based on space syntax and CFD simulation. This is an important work; well written and structured with only few literature review and comprehensive analyses. In general, this is an excellent contribution to knowledge in the respective field. The authors should however try to bring in more related literature from around the world to enrich their study and literature review.

Reviewer #2: This paper performed urban wind performance to improve comfort conditions in a cluster building block using CFD simulation. An optimization scenario was proposed to improve wind comfort conditions in winter conditions. This research has originality, but literature reviews need more work. In addition, the research method needs some clarifications, and significant concerns should be addressed before publication; these include:

1) What does it mean when you say, ‘To optimize the wind environment of the block more accurately, it analyzed, and 8 evaluated the accessibility of walking space of the block by relying on space syntax.’ Could you please clarify or explain how the space syntax analysis could provide accurate results for the wind simulation?

2) Number of references needs to be increased. It requires more work on reviewed papers regarding pedestrian wind comfort and urban airflow simulation.

3) The authors must provide detailed information on input data and boundary condition setting in the CFD model.

4) Even though the study calculated the residual errors of the CFD modeling, it is necessary to present validation results between the CFD modeling and field experiment.

5) More information about the site selection and site characteristic are required. Do you have any site selection criteria?

6) What research gap did this study observe from previous works? How do the study’s findings provide a significant contribution to the readers? Or any new insights will be provided in this study?

7) Please clarify why the study used space syntax and the airflow simulation model. It is necessary to present more fundamental concepts and numerical methods of the space syntax software. The parameter setting in space syntax should also be presented in this paper.

8) The characteristics of the optimized solutions compared to the base case need to be clarified. It would be better if the paper gave more explanations, pictures, or 2D/3D models representing before and after scenarios.

9) The paper needs more discussions on how the study's findings are similar or in contrast to similar studies

10) The paper needs to address the study’s limitations and recommendations for future studies.

11) Please check the format of in-text citations and other typos.

Reviewer #3: The authors have addressed an interesting topic which is essential for the future development of built environment. However, there is a clear lack of novelty in the paper and most of the analysis are usually performed before any regular construction process. I recommend the rejection of the article.

Major Issues:

1. Novelty: The authors have mentioned optimization of the block in the abstract. However, the section discussing on optimization analysis does not perform any of the commonly used statistical methods used for wind analysis in the built environment. Looking at the qualitative patterns of the wind profile does not fit the concept of optimization analysis.

2. ABL Homogeneity: The residual from the software is shown to demonstrate ABL homogeneity analysis. This is an inaccurate method. The authors need to follow the standard methodology used in the literature to perform homogeneity analysis by performing the fetch test and measuring the wind and turbulence profile at the inlet and outlet.

3. Grid Refinement Studies: Authors have not performed any grid refinement studies to show the sensitivity of the results to the grid resolution.

Other Issues:

There are a lot of figures included in the paper which can be dropped.

1. Figure 3 and 4 not required as there is no discussion of temperature.

2. Table 1 is not required as it is available in the literature

3. Figure 7 and 8 has no relevance to the paper. Can be removed.

4. Figure 10 is not essential.

5. Figure 11 should be removed and replaced with proper ABL homogeneity study.

6. There is no Figure 14 and jumps to Figure 15

7. Figures 17-19 are not required as it is common-sense knowledge

8. Authors need to improve the resolution of the contour plots.

6. PLOS authors have the option to publish the peer review history of their article (what does this mean?). If published, this will include your full peer review and any attached files.

Reviewer #1: No

Reviewer #2: **Yes: **Daranee Jareemit

Reviewer #3: No

---

## [Author Response · Author response to Decision Letter 0]

3 Nov 2023

1) What does it mean when you say, ‘To optimize the wind environment of the block more accurately, it analyzed, and 8 evaluated the accessibility of walking space of the block by relying on space syntax.’ Could you please clarify or explain how the space syntax analysis could provide accurate results for the wind simulation?

Response: As the outdoor space of the block is too large, the wind environment optimization cannot take into account every space, and the low accessibility space with less pedestrian reach frequency will bring less benefit to the optimization. Therefore, this paper mainly focuses on the pedestrian wind environment. In order to maximize the quality of the pedestrian wind environment of the pedestrian space, the paper analyzes the pedestrian accessibility of the pedestrian space based on space syntax. The highly accessible Spaces with the highest pedestrian travel frequency are delineated, and the shape characteristics and wind environment of these Spaces are accurately analyzed and optimized to ensure the effectiveness of wind environment optimization.

2) Number of references needs to be increased. It requires more work on reviewed papers regarding pedestrian wind comfort and urban airflow simulation.

Response: In the introduction part, the paper increases the number of citations of research achievements in the field of pedestrian wind environment simulation, and summarizes them, which provides the basis for the research method of the paper.

3) The authors must provide detailed information on input data and boundary condition setting in the CFD model.

Response: The authors add detailed information about the input data of the model and the setting of boundary conditions during the CFD simulation. The result is shown in Figures 6 and 7.

4) Even though the study calculated the residual errors of the CFD modeling, it is necessary to present validation results between the CFD modeling and field experiment.

Response: In order to ensure the rationality and feasibility of the simulation software and the selected simulation process, a comparison chart between the average field measured value and the simulated value has been added in the "Verification of simulation results" section, and specific data has been provided. Finally, it can be seen from the comparison chart that the curve changes have a relatively good agreement. It can reflect the change trend of actual wind speed, so it can prove the rationality and feasibility of the selection of simulation software and simulation process. The result is shown in Figure 9.

5) More information about the site selection and site characteristic are required. Do you have any site selection criteria?

Response: Lanzhou is located in a cold region in northwest China. Due to its geographical and climatic characteristics, urban blocks are usually enclosed. However, with the development of urbanization, the summer heat island effect of Lanzhou is constantly enhanced. The wind environment of enclosed blocks needs to be improved urgently. Therefore, the site selection standard of this study is that the block layout is enclosed and the flow of people is large, which is convenient to divide the high-accessibility area.

6) What research gap did this study observe from previous works? How do the study’s findings provide a significant contribution to the readers? Or any new insights will be provided in this study?

Response: The study and optimization of wind environment usually do not take into account the real needs of pedestrians. Some Spaces with good wind environment but few people can't really serve people. In this paper, space syntax is used to delineate highly accessible space areas, and the correlation analysis and improvement of wind environment and architectural form of these Spaces are carried out. In addition, the combination of space syntax and numerical simulation technology is also an innovation in the research methods and ideas of urban wind environment.

7) Please clarify why the study used space syntax and the airflow simulation model. It is necessary to present more fundamental concepts and numerical methods of the space syntax software. The parameter setting in space syntax should also be presented in this paper.

Response: The reasons for studying the use of space syntax and air flow simulation models are mentioned in the introduction. A large number of previous research results have proved that space syntax is a theory and method to study the relationship between spatial organization and human society through the quantitative description of human settlement spatial structure including buildings, settlements, cities and even landscapes, and has a better measurement of spatial accessibility. In addition, PHOENICS software and air flow simulation model are applicable to block scale. It can accurately reflect the wind environment at pedestrian height.

---

## [Decision Letter · Decision Letter 1]

23 Nov 2023

PONE-D-23-16872R1Evaluation and optimization of outdoor wind environment in block based on space syntax and CFD simulationPLOS ONE

Dear Dr. Li,

Thank you for submitting your manuscript to PLOS ONE. After careful consideration, we feel that it has merit but does not fully meet PLOS ONE’s publication criteria as it currently stands. Therefore, we invite you to submit a revised version of the manuscript that addresses the points raised during the review process.

The authors are suggested to consider the following points (which were also raised on the original submission) in their revised paper. Novelty: The authors have mentioned optimization of the block in the abstract. However, the section discussing on optimization analysis does not perform any of the commonly used statistical methods used for wind analysis in the built environment. Looking at the qualitative patterns of the wind profile does not fit the concept of optimization analysis.

2. ABL Homogeneity: The residual from the software is shown to demonstrate ABL homogeneity analysis. This is an inaccurate method. The authors need to follow the standard methodology used in the literature to perform homogeneity analysis by performing the fetch test and measuring the wind and turbulence profile at the inlet and outlet.

3. Grid Refinement Studies: Authors have not performed any grid refinement studies to show the sensitivity of the results to the grid resolution.

Other Issues:

There are a lot of figures included in the paper which can be dropped.

1. Figure 3 and 4 not required as there is no discussion of temperature.

2. Table 1 is not required as it is available in the literature

3. Figure 7 and 8 has no relevance to the paper. Can be removed.

4. Figure 10 is not essential.

5. Figure 11 should be removed and replaced with proper ABL homogeneity study.

6. There is no Figure 14 and jumps to Figure 15

7. Figures 17-19 are not required as it is common-sense knowledge

8. Authors need to improve the resolution of the contour plots. Please submit your revised manuscript by Jan 07 2024 11:59PM. If you will need more time than this to complete your revisions, please reply to this message or contact the journal office at plosone@plos.org. Please include the following items when submitting your revised manuscript:A rebuttal letter that responds to each point raised by the academic editor and reviewer(s). You should upload this letter as a separate file labeled 'Response to Reviewers'.A marked-up copy of your manuscript that highlights changes made to the original version. You should upload this as a separate file labeled 'Revised Manuscript with Track Changes'.An unmarked version of your revised paper without tracked changes. You should upload this as a separate file labeled 'Manuscript'.If applicable, we recommend that you deposit your laboratory protocols in protocols.io to enhance the reproducibility of your results. Protocols.io assigns your protocol its own identifier (DOI) so that it can be cited independently in the future. For instructions see: https://journals.plos.org/plosone/s/submission-guidelines#loc-laboratory-protocols. Additionally, PLOS ONE offers an option for publishing peer-reviewed Lab Protocol articles, which describe protocols hosted on protocols.io. Read more information on sharing protocols at https://plos.org/protocols?utm_medium=editorial-email&utm_source=authorletters&utm_campaign=protocols.

We look forward to receiving your revised manuscript.

Kind regards,

Muhammad Shakaib, PhD

Academic Editor

PLOS ONE

Journal Requirements:

Additional Editor Comments:

Comments of one of the reviewers are not addressed in the revised paper.

Reviewers' comments:

Reviewer's Responses to Questions

**Comments to the Author**

1. If the authors have adequately addressed your comments raised in a previous round of review and you feel that this manuscript is now acceptable for publication, you may indicate that here to bypass the “Comments to the Author” section, enter your conflict of interest statement in the “Confidential to Editor” section, and submit your "Accept" recommendation.

Reviewer #1: All comments have been addressed

Reviewer #2: All comments have been addressed

Reviewer #3: (No Response)

2. Is the manuscript technically sound, and do the data support the conclusions?

Reviewer #1: Yes

Reviewer #2: Yes

Reviewer #3: Partly

3. Has the statistical analysis been performed appropriately and rigorously? 

Reviewer #1: Yes

Reviewer #2: Yes

Reviewer #3: No

4. Have the authors made all data underlying the findings in their manuscript fully available?

Reviewer #1: Yes

Reviewer #2: Yes

Reviewer #3: No

5. Is the manuscript presented in an intelligible fashion and written in standard English?

Reviewer #1: Yes

Reviewer #2: Yes

Reviewer #3: Yes

6. Review Comments to the Author

Reviewer #1: The revisions completed accordingly with reviewers’ comments, which has now significantly improved the paper, presenting an innovative material on the subject, and therefore I would like to thank the authors for their contribution.

Reviewer #2: The authors have diligently addressed the concerns and recommendations according to my previous review.

Reviewer #3: Authors have not identified any of the concerns raised in the previous review and I recommend the rejection of the article.

7. PLOS authors have the option to publish the peer review history of their article (what does this mean?). If published, this will include your full peer review and any attached files.

Reviewer #1: No

Reviewer #2: No

Reviewer #3: No

---

## [Author Response · Author response to Decision Letter 1]

8 Jan 2024

1.Novelty: The authors have mentioned optimization of the block in the abstract. However, the section discussing on optimization analysis does not perform any of the commonly used statistical methods used for wind analysis in the built environment. Looking at the qualitative patterns of the wind profile does not fit the concept of optimization analysis.

Response: According to the suggestions of experts and scholars, Photoshop is used to process the air age cloud image in the section of wind environment optimization analysis. That is, the air age value and area represented by different color blocks are calculated, and the weighted average method is used to calculate the average air age of the study area. Combined with the practical application of this paper, the intermediate wind speed ratio is used as the relative comfort zone wind speed ratio RS to judge the area size that meets the comfort level of outdoor pedestrians at a height of 1.5m.

2. ABL Homogeneity: The residual from the software is shown to demonstrate ABL homogeneity analysis. This is an inaccurate method. The authors need to follow the standard methodology used in the literature to perform homogeneity analysis by performing the fetch test and measuring the wind and turbulence profile at the inlet and outlet.

Response: According to the suggestion, the horizontal homogeneity test of flow field is studied.

The atmospheric boundary layer should maintain horizontal homogeneity between upstream and downstream, so the inflow characteristics of computational fluid dynamics (CFD) simulations should be stable throughout the computational domain. In this study, the CFD numerical simulation of the wind tunnel test flow was carried out in a blank computational domain, and the wind speed flow characteristics at the center were extracted and compared with the inflow characteristics. As shown in Figures 5 , the CFD model established in this study basically meets the requirements of horizontal homogeneity. The wind speed characteristic curves of the inlet, center, and outlet are basically consistent.

3. Grid Refinement Studies: Authors have not performed any grid refinement studies to show the sensitivity of the results to the grid resolution.

Response: In the chapter of computational grid division, the grid form and precision used in the simulation of block wind environment are expounded. The specific operations are shown in Figure 4.

Other Issues:

There are a lot of figures included in the paper which can be dropped.

1. Figure 3 and 4 not required as there is no discussion of temperature.

Response: On the recommendation of the review experts, this image has been removed.

2. Table 1 is not required as it is available in the literature

Response: On the recommendation of the review experts, the table has been removed.

3. Figure 7 and 8 has no relevance to the paper. Can be removed.

Response: On the recommendation of the review experts, this image has been removed.

4. Figure 10 is not essential.

Response: On the recommendation of the review experts, this image has been removed.

5. Figure 11 should be removed and replaced with proper ABL homogeneity study.

Response: On the recommendation of the review experts, this image has been removed.

6. There is no Figure 14 and jumps to Figure 15

Response: The content has been revised according to the recommendations of the review experts.

7. Figures 17-19 are not required as it is common-sense knowledge

Response: On the recommendation of the review experts, this image has been removed.

8. Authors need to improve the resolution of the contour plots.

Response: The content has been revised according to the recommendations of the review experts.

---

## [Editor Report · Decision Letter 2]

11 Jan 2024

Evaluation and optimization of outdoor wind environment in block based on space syntax and CFD simulation

PONE-D-23-16872R2

Dear Dr. Li,

We’re pleased to inform you that your manuscript has been judged scientifically suitable for publication and will be formally accepted for publication once it meets all outstanding technical requirements.

Kind regards,

Muhammad Shakaib, PhD

Academic Editor

PLOS ONE
---

## [Editor Report · Acceptance letter]

20 Mar 2024

PONE-D-23-16872R2 

PLOS ONE

Dear Dr. Li, 

I'm pleased to inform you that your manuscript has been deemed suitable for publication in PLOS ONE. Congratulations! Your manuscript is now being handed over to our production team.

Kind regards, 

on behalf of

Dr. Muhammad Shakaib 

Academic Editor

PLOS ONE